# A Comprehensive Comparative Analysis and Phylogenetic Investigation of the Chloroplast Genome Sequences in Four *Astragalus* Species

**DOI:** 10.3390/cimb47120978

**Published:** 2025-11-25

**Authors:** Hai-Tao Ma, Qi-Yin Chen, Jie-Hua Rao, Kai-Ling Wang, Bei Jiang, Yong-Zeng Zhang

**Affiliations:** 1Yunnan Key Laboratory of Screening and Research on Anti-pathogenic Plant Resources from Western Yunnan, Dali 671000, China; pluto_0208@outlook.com (H.-T.M.);; 2College of Pharmacy, Dali University, Dali 671000, China

**Keywords:** *Astragalus*, complete chloroplast genome, genome comparison, phylogenetic analysis, IR lacking

## Abstract

*Astragalus* L. (Fabaceae), the largest plant genus with significant medicinal value, faces critical endangerment of its wild resources and a scarcity of chloroplast genomic data. We sequenced and assembled the complete chloroplast (cp) genomes of four *Astragalus* species (*A. yunnanensis*, *A. yunnanensis* subsp. *incanus, A. polycladus* and *A. polycladus* var. *nigrescens*) and performed comparative analyses with five previously published chloroplast genomes. The cp genomes of the four *Astragalus* species ranged in size from 122,868 bp to 125,752 bp, all lacking one inverted repeat (IR) region, thus belonging to the inverted repeat lacking clade (IRLC). Annotation revealed that each genome contained 110 unique genes, including 76 protein-coding genes, 30 tRNA genes, and 4 rRNA genes. Nucleotide diversity (Pi) analysis identified mutation hotspots, including 5 non-coding regions and 5 coding regions, which could serve as potential molecular markers. Additionally, evidence of positive selection was detected in 11 genes, suggesting their possible roles in adaptive evolution to environmental changes. Phylogenetic analysis revealed distinct clades, with *Astragalus* forming a monophyletic group within Fabaceae. Notably, closely related species, subspecies, and varieties were observed to cluster together, forming sister taxa. However, despite the conservation in cp genomes, *A. yunnanensis* and *A. yunnanensis* subsp. *incanus* exhibit significant morphological differentiation in leaf shape, leaf indumentum, and stem color. This paradox strongly suggests a markedly higher evolutionary rate in the nuclear genome compared to the chloroplast genome. The cp genomes of *Astragalus* presented here serve as a key resource for studying the genus’s genetic diversity and will aid in elucidating its intrageneric phylogeny.

## 1. Introduction

The *Astragalus* Linn. (Fabaceae, Papilionoideae, Galegeae) comprises approximately 3000 species, making it the largest genus of flowering plants, as well as the most species-rich vascular plant [1,2]. This taxon includes both annual and perennial species [3], with a distribution primarily concentrated in cold, arid continental regions of the Northern Hemisphere and South America, while being relatively rare in North America and Oceania [4,5]. In China, the genus is represented by 401 species, including 221 endemic species, predominantly distributed across northern and southern provinces, with particularly high diversity in Tibet (Himalayan region), Central Asia, and Northeast China [3]. *Astragalus* species contain various bioactive compounds including flavonoids, saponins, polysaccharides, amino acids, and trace elements [6,7,8]. These chemical constituent exhibit significant medicinal properties, demonstrating immunomodulatory, antitumor, antioxidant, hypoglycemic, hepatoprotective, and diuretic effects [9,10,11].

*Astragalus* exhibits characteristic papilionaceous floral structures with unique morphological synapomorphies [12], making its taxonomic delineation particularly challenging. Traditional morphological studies have demonstrated limited resolution in classifying infrageneric taxa and determining species boundaries. Recent advances in sequencing technologies have facilitated extensive research on plant chloroplast genomes, which in higher plants typically form circular quadripartite structures comprising a large single-copy region (LSC), a small single-copy region (SSC), and two intervening inverted repeat regions (IRs). These plastomes, generally spanning 120–170 kb, encode approximately 130 genes primarily involved in photosynthesis and chloroplast replication [13,14,15]. Due to their relatively small size, evolutionary conservation, and slow nucleotide substitution rates, chloroplast genomes have been widely employed as valuable tools for plant identification, evolutionary biology studies, and genetic diversity assessments [16]. However, within the Papilionoideae subfamily of Fabaceae, a distinct clade designated as the inverted repeat lacking clade (IRLC) has been identified [17,18], which has undergone extensive plastome rearrangements. Previous studies have documented various plastid genomic rearrangements in Fabaceae, including a 50 kb inversion present in most Papilionoideae species [19,20,21], loss of one IR copy [4,19,20,22], and deletions of the *infA*, *rpl22*, and *rps16* genes [23,24], as well as loss of *clpP* and *rpl2* introns [21,25,26]. Consequently, comprehensive investigation of complete chloroplast genome rearrangements and phylogenetic relationships within the *Astragalus* lineage is imperative to enhance our understanding of chloroplast evolution in Papilionoideae and Fabaceae as a whole.

To date, plastid genomes of about 38 *Astragalus* species (including 25 species of Neo-*Astragalus* (the New World aneuploid species) [4,27] and species belonging to other clades) have been deposited in NCBI (the National Center for Biotechnology Information). Despite significant advances in the genomics of *Astragalus* species, the increasing demand for *Astragalus* in recent years has led to the near depletion of wild *Astragalus* resources [28]. To better conserve *Astragalus* genetic resources, we sequenced the complete chloroplast genomes of four *Astragalus* species and conducted detailed comparative genomic and phylogenetic analyses with five previously reported *Astragalus* chloroplast genomes, as well as other IRLC plastomes (Figure 1). This study advances our understanding of chloroplast genome evolution within *Astragalus* and related species of the Fabaceae. Moreover, it offers invaluable genomic resources that can be instrumental for future conservation initiatives.

## 2. Materials and Methods

### 2.1. Plant Material Sampling, DNA Extraction and Sequencing

For this study, fresh leaf samples of four wild *Astragalus* species were collected and subsequently preserved in silica gel: *A. yunnanensis* and *A. yunnanensis* subsp. *incanus* were obtained from Baima Snow Mountain, Deqen County (28.46° N, 99.03° E), Diqing Tibetan Autonomous Prefecture, Yunnan Province, China, while *A. polycladus* and *A. polycladus* var. *nigrescens* were collected from Geza Township (28.08° N, 99.81° E), Shangri-La City, within the same prefecture. All specimens were authenticated by Professor Yongzeng Zhang and deposited at the Herbarium of Yunnan Key Laboratory of Screening and Research on Anti-pathogenic Plant Resources from Western Yunnan, with voucher codes 20240821-1, 20240821-2, 20240716-3, and 20240717-4, respectively. Genomic DNA was extracted using a modified CTAB protocol [29], with quality and concentration assessed through 1% agarose gel electrophoresis and spectrophotometry (Bio-Rad, Hercules, CA, USA). The DNA was sheared to approximately 350 bp fragments for library preparation. Sequencing was performed on the DNBSEQ-T7 platform, followed by quality filtering using fastp v.0.23.2 [30] to remove low-quality reads. Sequencing depth was evaluated using Samtools v1.17 [31]. The entire sequencing process was conducted by Wuhan Benagen Tech Co., Ltd. (Wuhan, China).

### 2.2. Chloroplast Genome Assembly and Annotation

The high-quality clean reads were assembled de novo using GetOrganelle software v1.7.5 [32] with default parameters for plant chloroplast genome reconstruction. The assembled chloroplast genomes were annotated using CPGAVAS2 (http://www.herbalgenomics.org/cpgavas2, accessed on 16 February 2025) [33], with graphical maps generated by OGDRAW v1.3.1 [34]. tRNA genes were identified through tRNAscan-SE v2.0.9 [35], while rRNA genes were annotated via BLASTN v2.8.1 [36]. Annotation errors were manually corrected using CPGView (http://www.1kmpg.cn/cpgview, accessed on 16 February 2025) [37] and Apollo [38]. The fully annotated chloroplast genome sequences of the four *Astragalus* species have been deposited in the NCBI GenBank database under accession numbers PV156652.1, PV156653.1, PV910878.1, and PV910879.1.

### 2.3. Codon Usage Bias Analysis

To mitigate sampling bias, protein-coding regions longer than 300 bp were exclusively analyzed. All coding sequences shorter than 300 bp were systematically excluded to ensure robust codon usage pattern determination. The relative synonymous codon usage (RSCU) value, defined as the ratio of observed codon frequency to expected frequency under equal usage, served as a reliable indicator of codon preference [39]. RSCU values for the nine *Astragalus* chloroplast genomes were computed using CodonW 1.4.4 (codon table = 11) [40], following standard bioinformatic protocols for plastid genome analysis.

### 2.4. Repeat Element and SSR Analysis

The chloroplast genomes of nine *Astragalus* species were analyzed for dispersed repeats, tandem repeats, and simple sequence repeats (SSRs) using specialized bioinformatics tools. Dispersed repeats (including forward, reverse, complementary, and palindromic types) were identified using REPuter (https://bibiserv.cebitec.uni-bielefeld.de/reputer, accessed on 12 March 2025) [41] with the following parameters: minimum repeat length = 30 bp, Hamming distance = 3, and maximum computed repeat size = 5000 bp. Tandem repeats were detected using the online Tandem Repeats Finder program [42] with default settings. SSR analysis was performed with MISA (https://webblast.ipk-gatersleben.de/misa/, accessed on 12 March 2025) [43], applying minimum repeat unit thresholds of 8, 4, 4, 3, 3, and 3 for mono-, di-, tri-, tetra-, penta-, and hexa-nucleotide motifs, respectively.

### 2.5. Comparative Genome and Sequence Divergence Analyses

Comparative analysis of the nine *Astragalus* chloroplast genomes was performed using a suite of bioinformatics tools. Complete plastome alignments were initially conducted using MAFFT v.7.313 [44], followed by extraction of consensus coding and intergenic regions. Nucleotide diversity (Pi) was calculated through sliding-window analysis (window length = 600 bp, step size = 200 bp) in DnaSP v.6.0 [45] based on the alignment results. Genomic divergence and mutation hotspots were identified using the online mVISTA platform (http://genome.lbl.gov/vista/index.shtml, accessed on 20 March 2025) in Shuffle-LAGAN mode, with the annotated *A. yunnanensis* chloroplast genome serving as the reference sequence [46].

### 2.6. Analysis of Synonymous (Ks) and Non-Synonymous (Ka) Substitution Rate

To elucidate the role of natural selection in shaping the molecular evolution of *Astragalus* chloroplast genomes, we employed synonymous (Ks) and nonsynonymous (Ka) substitution rates along with their ratio (Ka/Ks). All protein-coding genes were aligned using MAFFT v.7.313 [44], followed by calculation of Ks, Ka, and Ka/Ks values through KaKs_Calculator 2.0 [47]. The selection pressure was interpreted as follows: Ka/Ks > 1 indicates positive selection, Ka/Ks = 1 suggests neutral evolution, and Ka/Ks < 1 signifies purifying selection.

### 2.7. Phylogenetic Analysis

To investigate the phylogenetic relationships within *Astragalus* and the IRLC of Fabaceae, we retrieved complete chloroplast genomes of 49 species from the NCBI database, supplemented with four *Astragalus* species from this study. *Lotus corniculatus* and *L. corniculatus* subsp. *japonicus* were selected as outgroups for phylogenetic reconstruction. The sampling strategy for selecting these 53 taxa from the IRLC was designed with the following objectives: (1) to include representative species from all major genera within the IRLC for which complete chloroplast genome sequences are available; (2) to specifically oversample genera phylogenetically close to *Astragalus* (such as *Oxytropis*, *Caragana*, and *Glycyrrhiza*) to robustly test the monophyly and phylogenetic position of *Astragalus*; and (3) to base our selection on all available, high-quality, and complete chloroplast genome sequences in public databases (e.g., NCBI). All sequences were aligned using MAFFT v.7.313 [44] and subsequently trimmed with Trimal v1.4 [48]. The optimal substitution models were determined through ModelFinder [49] in PhyloSuite v1.2.3 [50], identifying GTR+G4+F for maximum likelihood (ML) analysis and GTR+G+F for Bayesian inference (BI). ML trees were constructed with IQ-tree 2.2.0 [51] using 1000 bootstrap replicates to assess branch support [52]. Bayesian analysis was performed in MrBayes v3.2.7 [53] with two parallel runs of 1,000,000 generations, discarding the initial 25% as burn-in. Final tree visualization was conducted using FigTree v1.4.4 [54].

## 3. Results

### 3.1. Chloroplast Genome Features of Astragalus Species

This study analyzed nine cp genomes, including four newly sequenced genomes (Appendix A) and five previously published ones. The four newly sequenced *Astragalus* cp genomes ranged in size from 122,868 bp to 125,752 bp, with GC contents between 34.14% and 34.22% (Table 1), demonstrating highly conserved plastome architectures. Due to the absence of inverted repeat (IR) regions, they lack the typical quadripartite structure found in most angiosperm chloroplast genomes, exhibiting instead the characteristic IRLC organization with correspondingly shorter lengths (Figure 2). Average read depths for gene coverage were 1454.51×, 2773.59×, 4696.17×, and 4570.70×, respectively (Appendix A). Notably, the cp genomes of the four *Astragalus* species lack the *infA*, *rps16*, and *rpl22* genes, as well as the first intron of the *clpP* gene.

The annotation results of the cp genomes showed that all four *Astragalus* plastomes contained 110 genes, including 76 protein-coding genes (PCGs), 30 transfer RNA (tRNA) genes, and 4 ribosomal RNA (rRNA) genes (Figure 2; Table 2). Among the four *Astragalus* species, 10 PCGs (*ndhA*, *ndhB*, *petB*, *petD*, *atpF*, *rps12*, *rpl2*, *rpl16*, *rpoC1*, and *clpP*) and 6 tRNA genes (*trnA*-*UGC*, *trnG*-*UCC*, *trnI*-*GAU*, *trnK*-*UUU*, *trnL*-*UAA*, and *trnV*-*UAC*) contained 1 intron each, while *ycf3* had 2 introns (Appendix A). Additionally, cis-spliced and trans-spliced genes were identified in the four *Astragalus*, with a total of 10 PCGs (*ndhA*, *ndhB*, *rpl2*, *rpl16*, *petD*, *petB*, *clpP*, *atpF*, *rpoC1*, and *ycf3*) being cis-spliced genes, each containing 1–2 introns (Appendix A). The *rps12* gene underwent trans-splicing, with its 3′ end lacking an intron (Appendix A). All genes were classified into four categories: the first group comprised 44 photosynthesis-related genes; the second group included 57 self-replication-related genes; the third group contained 5 other functional genes; and the fourth group consisted of 4 unknown function genes (Table 2).

Comparative analysis of the four newly sequenced *Astragalus* chloroplast genomes with five previously published ones revealed total sequence lengths ranging from 122,796 to 125,752 base pairs. The absence of IRs resulted in the shortest chloroplast genome in *A. laxmannii* (122,796 bp), while *A. yunnanensis* exhibited the longest genome (125,752 bp) (Table 1). Notably, *A. membranaceus*, *A. laxmannii*, and *A. membranaceus* var. *mongholicus* displayed distinct gene counts (107, 106, and 109 genes, respectively) compared to the consistent 110 genes observed in the other six *Astragalus* species (Table 1). This variation primarily stemmed from differences in tRNA gene numbers, while PCGs and rRNA gene counts remained conserved across all examined genomes.

From the perspective of gene content, PCGs were the most abundant among the nine plant species, accounting for approximately half of the entire genome length. This was followed by tRNA genes, which were considerably shorter in length compared to other genes (Table 1). Overall, the chloroplast genome sequences of these nine *Astragalus* species exhibited highly similar lengths and gene compositions. We further analyzed the differences in GC content among these three gene categories. The rRNA genes displayed the highest GC content, consistently exceeding 50%. tRNA genes ranked second in GC content, while PCGs exhibited the lowest GC content, averaging around 36%. Additionally, the mean GC content across all nine species was approximately 34% (Table 1), suggesting a relatively conserved sequence evolution within the *Astragalus*.

### 3.2. Codon Usage Analysis

Analysis of codon usage patterns across nine *Astragalus* species revealed both conserved and species-specific trends. We identified 61 relative synonymous codon usage (RSCU) values in *Astragalus* plastomes, with total codon counts ranging from 18,018 (*A. arpilobus*) to 20,206 (*A. yunnanensis*) (Appendix A). The species exhibited minimal variation in codon numbers per amino acid, maintaining consistent codon preference patterns. Among these codons, leucine (Leu) was the most abundant amino acid (10.51–10.66% of total occurrences), followed by isoleucine (Ile) (8.82–8.99%), while cysteine (Cys) was the rarest (1.06–1.09%) (Appendix A). Among the 61 codons, 29 showed RSCU values > 1, with leucine’s TTA demonstrating the highest RSCU (2.05–2.18), while 30 codons had RSCU values < 1. The methionine (Met) and tryptophan (Trp) codons ATG and TGG exhibited RSCU = 1, indicating no usage bias (Figure 3B; Appendix A). Notably, among codons with RSCU > 1, all except TTG (leucine) terminated with A/U (T) (Figure 3A; Appendix A), demonstrating A/U (T)-ending codon dominance in *Astragalus* chloroplast genomes.

### 3.3. Repeat Sequence and SSR Analyses

Comprehensive analysis of *Astragalus* chloroplast genomes revealed 386 tandem repeats (Appendix A), with *A. polycladus* (31 repeats) showing the lowest frequency and *A. yunnanensis* subsp. *incanus* (70 repeats) exhibiting the highest count (Figure 4A; Appendix A). While tandem repeat lengths varied across the nine plastomes, the majority (50–59 bp) clustered within a specific size range (Figure 4B). Four distinct long repeat types were identified: forward, reverse, complementary, and palindromic repeats (Figure 4A), with total counts ranging from 35 (*A. polycladus* var. *nigrescens*) to 335 (*A. yunnanensis* subsp. *incanus*) (Appendix A). Forward repeats predominated (15 [*A. polycladus* var. *nigrescens*] to 279 [*A. yunnanensis* subsp. *incanus*]), followed by palindromic repeats (15 [*A. tenuis*] to 27 [*A. yunnanensis* subsp. *incanus*]) (Figure 4A). Reverse and complementary repeats occurred less frequently, ranging from 2 (*A. polycladus* var. *nigrescens*, *A. polycladus*, *A. tenuis*, *A. arpilobus*) to 20 (*A. yunnanensis* subsp. *incanus*) and 1 (*A. tenuis*) to 9 (*A. yunnanensis* subsp. *incanus*), respectively (Figure 4A–D). Notably, 30–39 bp represented the most prevalent length category across all long repeat types (Figure 4B–D).

A comprehensive analysis of SSRs in *Astragalus* chloroplast genomes identified 242–265 SSRs across the nine species, with all specimens containing mono-, di-, tri-, tetra-, and penta-nucleotide repeats (Figure 4E; Appendix A). Notably, hexa-nucleotide repeats were exclusively detected in *A. yunnanensis*, *A. yunnanensis* subsp. *incanus*, and *A. tenuis*, being absent in the remaining six species (Figure 4E; Appendix A). The quantitative distribution of SSR types was as follows: mono-nucleotide (142–151), di-nucleotide (75–92), tri-nucleotide (7–13), tetra-nucleotide (9–14), penta-nucleotide (1–6), and hexa-nucleotide (0–4) repeats (Figure 4E; Appendix A). Mono- and di-nucleotide SSRs demonstrated particularly high prevalence across all sequenced genomes. The majority of mono-nucleotide repeats consisted of A/T bases with minimal G/C content, while AT/TA sequences dominated the di-nucleotide repeats, a pattern consistently observed in all nine species (Appendix A). Further analysis of SSR distribution between genic and intergenic regions revealed significantly lower SSR abundance in coding regions compared to intergenic spacers (Appendix A).

### 3.4. Sequence Divergence Analysis

To elucidate conserved and divergent characteristics within *Astragalus* species, we conducted comparative analyses of plastid sequences from four newly sequenced and five previously reported *Astragalus* taxa, using mVISTA with *A. yunnanensis* as the reference genome. The results demonstrated high similarity among all nine chloroplast genomes, while revealing sequence divergence primarily in intergenic spacer (IGS) regions, including *trnK*-*UUU*-*rbcL*, *rbcL*-*atpB*, *ndhJ*-*trnF*-*GAA*, *trnL*-*UAA*-*trnT*-*UGU*, *ycf3*-*psaA*, *trnG*-*GCC*-*psbZ*, *trnS*-*UGA*-*psbC*, *psbD*-*trnT*-*GGU*, *atpI*-*atpH*, *petA*-*psbJ*, *psbE*-*trnW*-*CCA*, *rpl14*-*rps8*, *rpl36*-*rps11*, *rps12*-*trnV*-*GAC*, *trnN*-*GUU*-*ycf1* and *rpl32*-*ndhF* (Figure 5). Protein-coding genes exhibited strong conservation, with notable exceptions in *rps4*, *ycf3*, *rpoC1*, *accD*, *rps18*, *clpP*, *rpl16*, *ycf2* and *ycf1*. Comparative analysis further revealed significantly higher sequence conservation in coding regions relative to non-coding regions.

### 3.5. Nucleic Acid Polymorphism Analysis

To assess sequence divergence patterns, we performed nucleotide diversity (Pi) analysis on both coding and intergenic regions across the nine *Astragalus* chloroplast genomes using DnaSP v6 software. The calculated Pi values ranged from 0.00000 to 0.15972, with a mean value of 0.01518 (Appendix A). The most variable intergenic regions were identified as *trnfM*-*CAU*-*trnG*-*GCC* (Pi = 0.05226), *atpI*-*atpH* (0.12194), *psbT*-*psbN* (0.10256), *trnI*-*CAU*-*ycf2* (0.05120), and *ndhI*-*ndhG* (0.05288), while the most polymorphic coding regions included *rpl20* (0.02531), *clpP* (0.03028), *trnV*-*GAC* (0.02855), *trnA*-*UGC* (0.15972), and *ycf1* (0.03290) (Figure 6). These highly variable regions represent potential molecular markers for *Astragalus* species identification.

### 3.6. Selective Pressure Analysis

Using *Astragalus yunnanensis* chloroplast genome as the reference, we calculated the Ka/Ks ratios for 76 PCGs across eight *Astragalus* species (Figure 7). The analysis revealed that most genes exhibited Ka/Ks ratios < 1, indicating strong purifying selection (Appendix A). Notably, *rps11* and *ycf1* demonstrated Ka/Ks > 1 across all examined species, while nine additional genes (*cemA*, *ndhB*, *rpl20*, *rpoA*, *rps18*, *rps2*, *rps3*, *rps7* and *ycf2*) showed Ka/Ks ratios > 1 in specific *Astragalus* lineages, providing evidence for positive selection acting on these genes during species diversification. The observed interspecific variation in Ka/Ks ratios among these eleven genes suggests their potential roles in adaptive evolution within the genus.

### 3.7. Phylogenetic Relationship Analysis

To elucidate the evolutionary relationships within *Astragalus* and related Papilionoideae taxa, we performed comprehensive phylogenetic analyses encompassing 53 species (Appendix A). Both maximum likelihood (ML) and Bayesian inference (BI) analyses yielded identical tree topologies, presented here as a single consensus tree (Figure 8A,B). The phylogenetic reconstruction revealed five major clades: (1) *Onobrychis*, *Vicia*, *Lathyrus*, *Hedysarum*, and *Oxytropis*; (2) *Cicer*, *Melilotus*, and *Trifolium*; (3) *Astragalus*; (4) *Caragana*; and (5) *Glycyrrhiza.* Notably, the newly sequenced *A. yunnanensis* showed close affinity with *A. yunnanensis* subsp. *incanus*, and *A. polycladus* clustered with *A. polycladus* var. *nigrescens*, demonstrating tight phylogenetic relationships among species, subspecies, and varieties within the genus.

## 4. Discussion

### 4.1. Chloroplast Genome Structure

In this study, we characterized the cp genomes of four *Astragalus* species and conducted comparative analyses with five previously reported congeners. Distinct from most angiosperms, both the newly sequenced and previously reported *Astragalus* plastomes exhibit a notable absence of IRs, resulting in ambiguous boundaries between the LSCs and SSCs. Notably, the cp genomes of the four *Astragalus* species were found to lack *infA*, *rps16*, and *rpl22* genes, along with the first intron of the *clpP* gene, which is consistent with previous reports in other congeneric species including *A. membranaceus*, *A. membranaceus* var. *mongholicus*, *A. iranicus* and *A. melilotoides* [55,56,57]. Among these missing elements, *infA* represents an exceptionally unstable chloroplast gene in flowering plants [24], *rps16* encodes a cruciform DNA-binding protein [58], and *rpl22* produces ribosomal protein CL22 [59]. Their absence likely reflects either functional transfer to the nucleus or replacement by nuclear genes of prokaryotic/eukaryotic origin, as documented in *Glycine max* (L.) Merr. (nuclear relocation of *infA*) [24] and *Medicago sativa* L. (mitochondrion-derived nuclear *rps16* substitution) [60]. These observations suggest potential nuclear translocation of *infA*, *rps16*, and *rpl22* in these *Astragalus* species. However, given the current scarcity of *Astragalus* chloroplast genome data, extensive experimental validation remains imperative to confirm these evolutionary inferences.

### 4.2. Characteristics of Codon Usage and Repetitive Sequences

Codon usage bias serves as a valuable indicator for investigating evolutionary history, predicting expression levels, and understanding molecular-level evolutionary processes acting on genomes [61,62]. Our analyses revealed that *Astragalus* species, like most plants, predominantly employ A/U-ending codons (RSCU > 1), with the exception of UUG [63]. This translational preference for A/U at the third codon position likely reflects the combined effects of natural selection and mutational bias during chloroplast genome evolution [64]. cp genomes are rich in SSRs, LRSs, and highly divergent regions—critical genetic markers closely associated with species origin and diversification [65]. The examined *Astragalus* plastomes contained 35–335 long repeats encompassing all four types (forward, palindromic, reverse, and complementary), though forward (F) and palindromic (P) repeats substantially outnumbered reverse (R) and complementary (C) types. SSRs numbered 242–265 per genome, with mononucleotide repeats being most abundant (56.98–58.68% of total SSRs), while penta- and hexa-nucleotide motifs were exceptionally rare. Notably, most SSRs exhibited AT-rich composition with minimal GC content, consistent with patterns observed in other Fabaceae species [56,57,66]. These SSR repositories provide valuable foundations for developing genetic markers applicable to species identification, phylogenetic reconstruction and ecological studies in *Astragalus*.

### 4.3. Comparative Genomic Analysis and Nucleotide Diversity

Plastid genomes harbor abundant nucleotide polymorphisms that serve as valuable DNA barcodes for elucidating interspecific and intergeneric relationships [67,68]. Our analyses identified 10 hypervariable regions with significantly elevated divergence values, including 5 intergenic spacers (*trnfM*-*CAU*-*trnG*-*GCC*, *atpI*-*atpH*, *psbT*-*psbN*, *trnI*-*CAU*-*ycf2* and *ndhI*-*ndhG*) and 5 coding regions (*rpl20*, *clpP*, *trnV*-*GAC*, *trnA*-*UGC* and *ycf1*). These highly polymorphic loci represent promising candidate DNA barcodes for phylogenetic reconstruction, species identification, and population genetic studies in *Astragalus*. Furthermore, mVISTA-based divergence analysis revealed substantially greater sequence variation in non-coding regions compared to coding sequences, suggesting that intergenic spacers are exceptionally well-suited for the development of molecular markers within this genus.

### 4.4. Analysis of Selection Pressure

The Ka and Ks nucleotide substitution patterns serve as crucial indicators of gene evolution [69]. Selection pressure on genes is reflected by the Ka/Ks ratio, where values < 1, =1, and >1 signify purifying selection, neutral evolution, and positive selection, respectively, with most PCGs typically undergoing purifying selection [70,71]. Among the 76 PCGs analyzed, only 11 exhibited Ka/Ks ratios > 1, indicating positive selection and rapid evolutionary adaptation. Notably, *rps11* and *ycf1* demonstrated Ka/Ks > 1 across all examined *Astragalus* species. The *rps11* gene, encoding a component of the 30S ribosomal subunit involved in chloroplast protein translation, may undergo positive selection to modulate translation rates in response to environmental constraints [72]. As one of the largest and most conserved chloroplast genes, *ycf1* likely stabilizes photosynthetic complexes, with its selective patterns potentially enhancing photosynthetic efficiency under low-light or hypoxic conditions [73]. The positive selection observed in these genes suggests their critical roles in regulating plastid gene expression and environmental adaptation.

### 4.5. Phylogenetic Relationships of IR-Lacking Clades

Due to its simple structure and maternal inheritance pattern, the chloroplast genome has been widely employed for resolving evolutionary relationships among species [74]. Our analysis of chloroplast genomic datasets from 53 species within the IRLC revealed that *Astragalus* forms a well-supported monophyletic cluster divided into two major clades, thereby clarifying the phylogenetic positions of the newly sequenced *Astragalus* species within the genus. While previous studies identified *Oxytropis* and *Caragana* as sister groups to *Astragalus* [13,75], our current analysis demonstrated closer affinity between *Oxytropis* and *Hedysarum*, with *Caragana* forming a distinct lineage. The *Astragalus*–*Oxytropis*–*Hedysarum* clade showed sister relationships with *Lathyrus*–*Vicia*–*Onobrychis* before associating with *Caragana*. We posit that this topological discrepancy primarily stems from differences in sampling strategies. First, increased taxonomic sampling enhances the statistical power of phylogenetic reconstruction by providing more comprehensive genetic variation data, thereby improving the accuracy of evolutionary inference and reducing topological uncertainty [76]. Second, expanded sampling mitigates potential biases and offers a more complete perspective of genetic diversity, consequently improving tree resolution and enabling clearer differentiation of closely related species or populations [77]. Previously, *A. polycladus* var. *nigricans* was merged into *A. polycladus*. Phylogenetic analysis revealed a high degree of similarity in their chloroplast genome sequences (bootstrap support ≥ 95%), providing genomic evidence to support their taxonomic treatment as a single species.

The highly conserved sequences of the chloroplast genomes among the four *Astragalus* species at the species level reaffirm the characteristically low evolutionary rate of this genome in plants and further support a very close phylogenetic relationship among these species, subspecies, and varieties. However, in stark contrast, *A. yunnanensis* and *A. yunnanensis* subsp. *incanus* exhibit significant morphological differentiation in leaf shape, leaf indumentum, and stem color. This paradox of “molecular conservation” versus “phenotypic divergence” strongly suggests that the nuclear genome has evolved at a much faster rate than the chloroplast genome [78,79], and the observed morphological differences are likely the result of adaptive evolution to different habitats or selective pressures [80].

The observed morphological divergence may originate from distinct regulatory pathways in the nuclear genome: the formation and variation in leaflet trichomes are primarily associated with the coordinated regulation of plant hormone signaling (e.g., GAs and CTKs) and specific transcription factors (e.g., MYB, bHLH). This conserved network may have been specifically modified in the genus *Astragalus* in response to adaptation to different habitats [81,82,83]. In contrast, stem color variation involves chlorophyll and anthocyanin metabolic pathways, with the functions of related genes (e.g., *APRR2*) and regulatory factors (e.g., GLKs, MYB) having been confirmed in multiple plant species [84,85]. These findings support an integrative hypothesis: natural selection may independently shape nuclear genes controlling trichome development and pigmentation, thereby facilitating the coordinated evolution of multiple traits and adaptive divergence. Therefore, future studies should focus on comparative genomic analyses of the nuclear genome, aiming to identify the key genetic loci underlying these critical phenotypic traits.

## 5. Conclusions

In this study, we sequenced and assembled the chloroplast genomes of four *Astragalus* species, confirming their classification within the IRLC through comparative analysis with five previously published plastomes. Our comprehensive molecular characterization encompassed codon usage patterns, repeat sequence distribution, hotspot region identification, selection pressure analysis, and phylogenomic assessment. We identified ten highly variable loci (5 intergenic: *trnfM-CAU*-*trnG-GCC*, *atpI*-*atpH*, *psbT*-*psbN*, *trnI-CAU*-*ycf2*, *ndhI*-*ndhG*; 5 coding: *rpl20*, *clpP*, *trnV-GAC*, *trnA-UGC*, *ycf1*) that represent promising molecular markers for species identification and phylogenetic studies, pending further validation. Positive selection signals were detected in 11 genes functionally associated with photosynthesis and related physiological processes, suggesting adaptive evolution to diverse environmental conditions in *Astragalus*. The reconstructed phylogenetic tree establishes a robust framework for species delineation, genetic diversity assessment, and evolutionary studies within the genus. These remarkable findings not only substantially expand the genomic resources available for *Astragalus* but also offer invaluable references for phylogenetic reconstruction and conservation initiatives. The study lays a substantial foundation for future evolutionary analyses, taxonomic revisions, and investigations of genetic diversity across broader IRLC lineages.

## Figures and Tables

**Figure 1 cimb-47-00978-f001:**
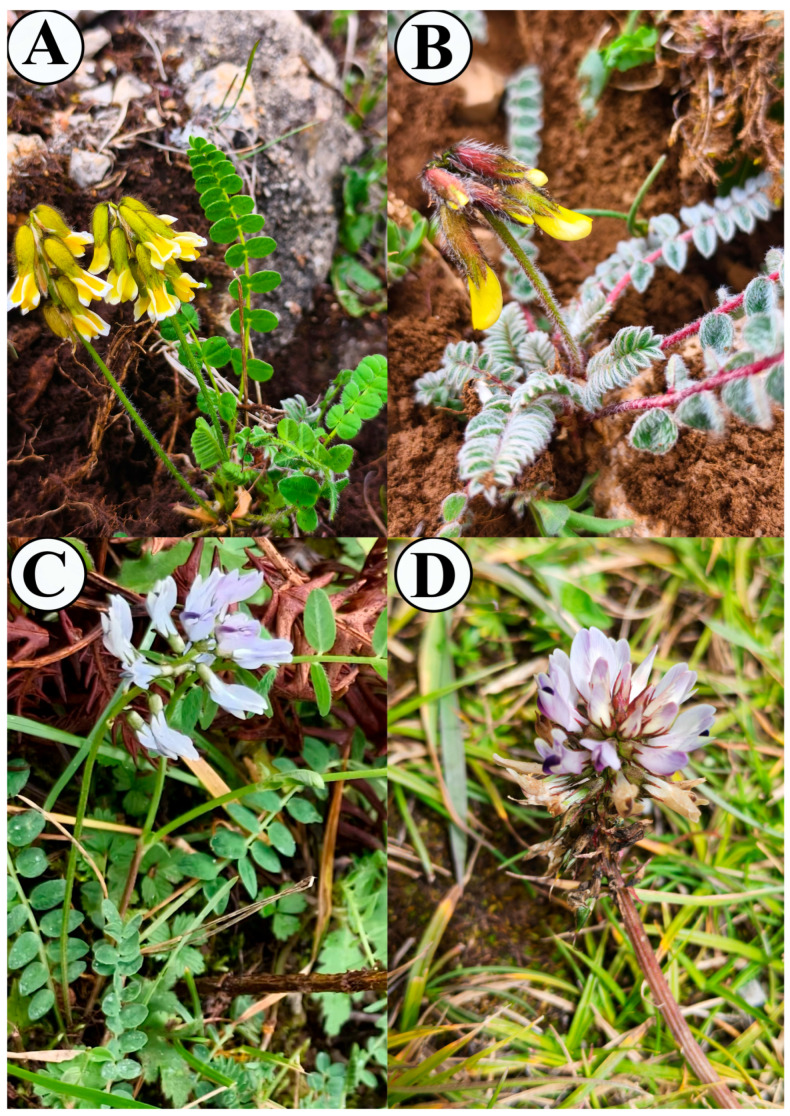
Plant morphology of *A. yunnanensis*, *A. yunnanensis* subsp. *incanus*, *A. polycladus*, and *A. polycladus* var. *nigrescens* (**A**–**D**).

**Figure 2 cimb-47-00978-f002:**
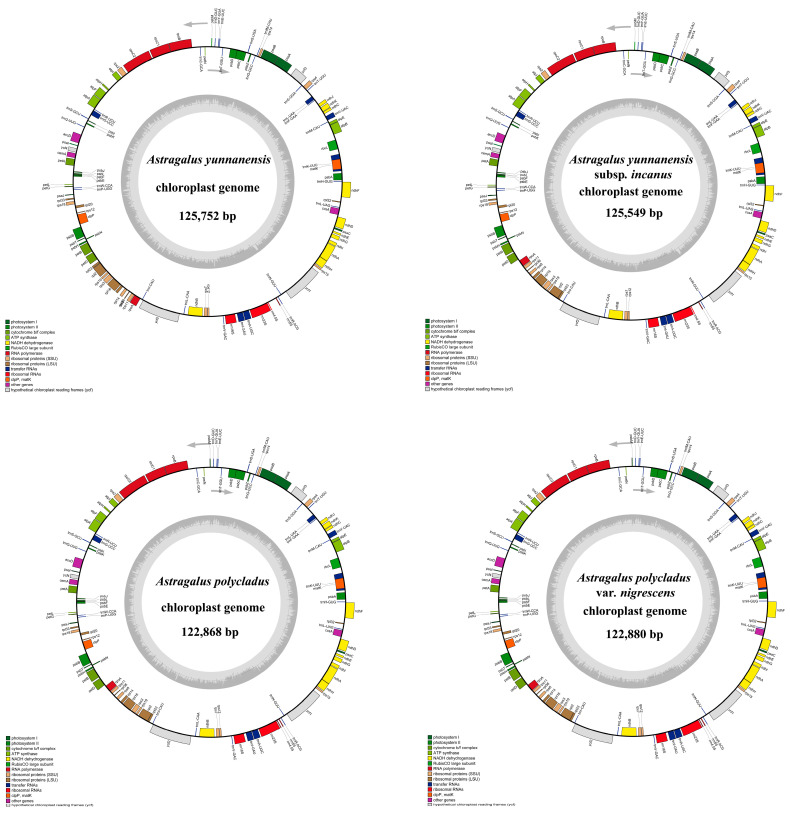
Chloroplast genome maps of *Astragalus* with annotated genes. Genes within the circle are clockwise, while those beyond the circle are counterclockwise. Different colors indicate functional gene groups. The darker and lighter shades of gray in the inner circle represent the content of GC and AT, respectively.

**Figure 3 cimb-47-00978-f003:**
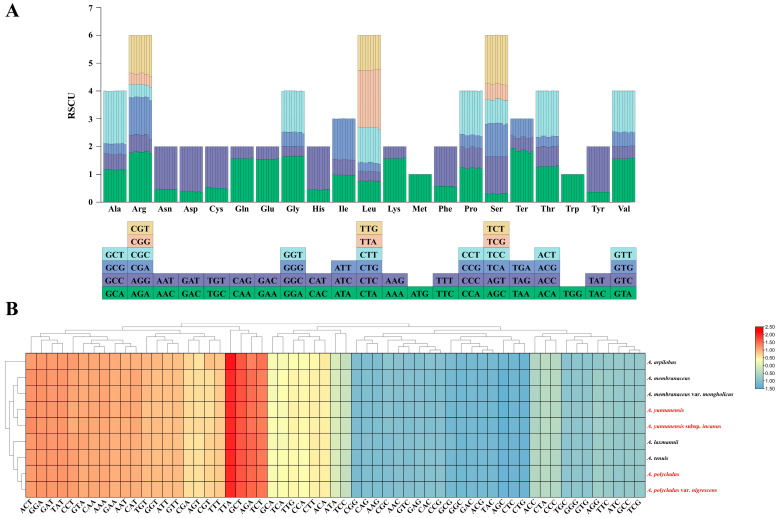
Codon bias analysis of 9 *Astragalus* species. (**A**) Relative Synonymous Codon Usage (RSCU) analysis. The *x*-axis shows the 20 standard amino acids, and the *y*-axis shows the corresponding RSCU values. (**B**) Heatmap analysis of the Relative Synonymous Codon Usage (RSCU) values for protein-coding genes. Red and blue colors indicate higher and lower RSCU values, respectively. Species labeled in red are those sequenced in this study.

**Figure 4 cimb-47-00978-f004:**
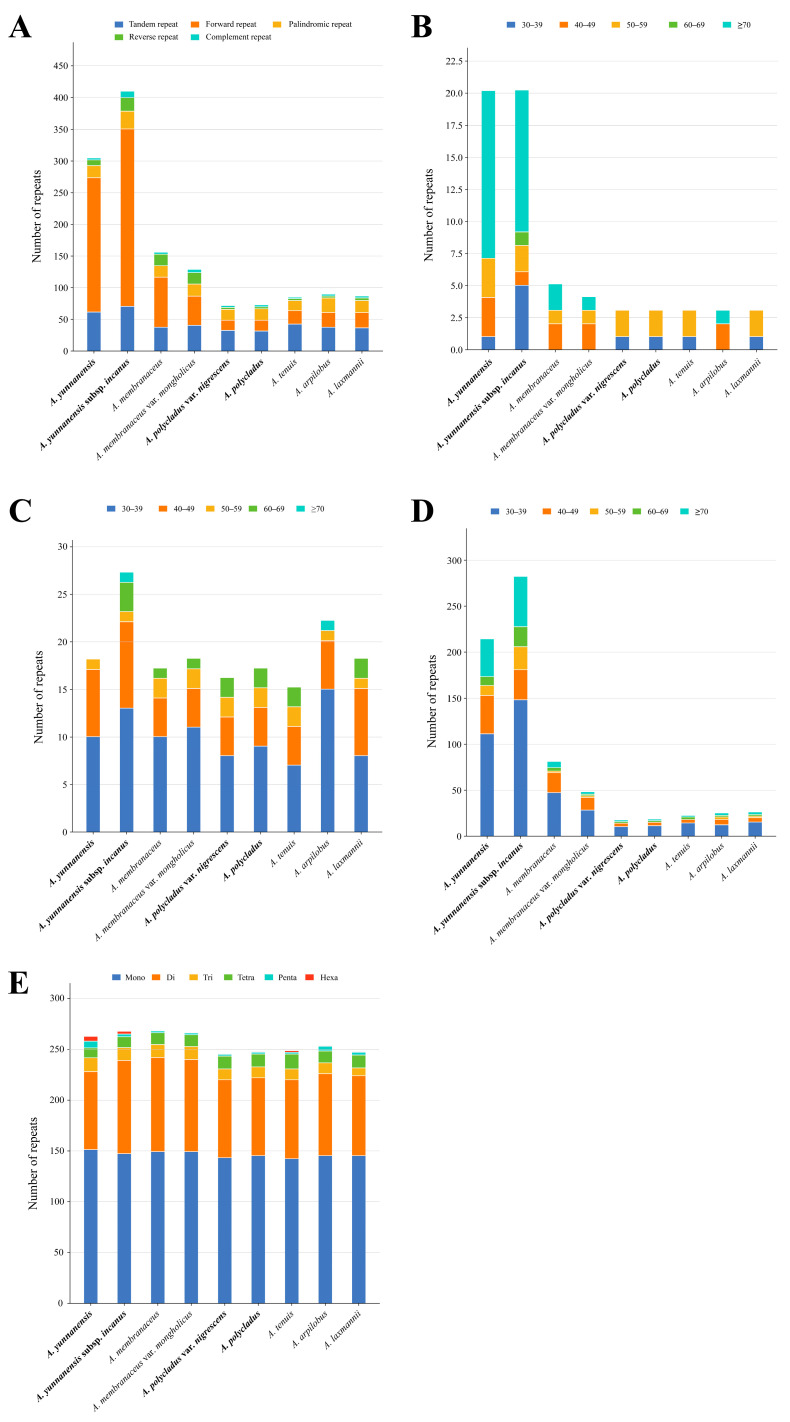
Analysis of repeats and SSRs in 9 complete chloroplast genomes of the *Astragalus*. (**A**) Different types of repeats in each chloroplast genome. (**B**) Numbers of tandem repeats more than 30 bp long in each chloroplast genome. (**C**) Numbers of palindromic repeats more than 30 bp long in each chloroplast genome. (**D**) Numbers of forward repeats more than 30 bp long in each chloroplast genome. (**E**) Total numbers and different types of SSRs detected in each chloroplast genome. Mono: mononucleotide, Di: dinucleotide, Tri: trinucleotides, Tetra: tetranucleotide, Penta: pentanucleotide, Hexa: hexanucleotide. The species in bold are sequenced in this study.

**Figure 5 cimb-47-00978-f005:**
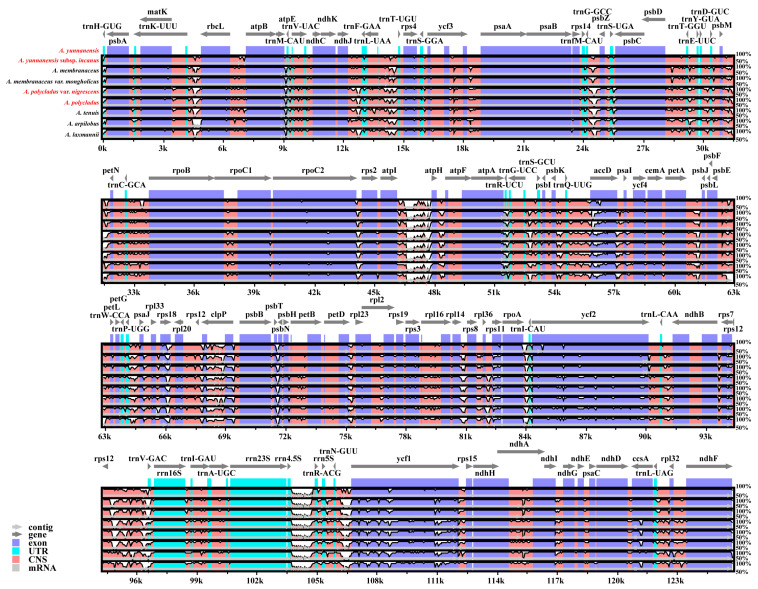
The chloroplast genome of nine *Astragalus* species were compared by mVISTA. The gray arrow in the figure indicates the direction of gene translation. The x-axis represents the coordinates in the chloroplast genome; the y-axis represents the percentage between 50 and 100%; Blue indicates protein coding (exon); light green indicates untranslated region (UTR); orange indicates conserved non-coding sequences (CNSs). Species labeled in red are those sequenced in this study.

**Figure 6 cimb-47-00978-f006:**
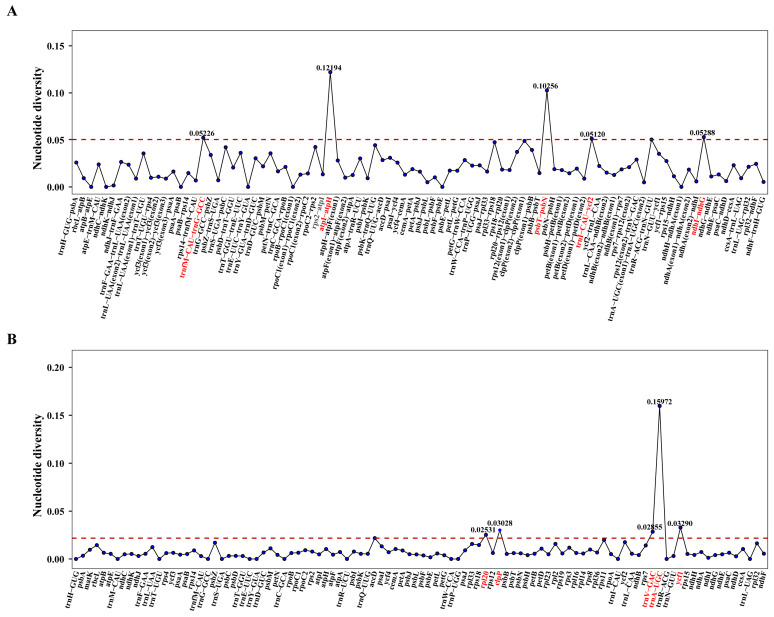
Nucleotide diversity (Pi) across (**A**) intergenic spacer regions (IGS) and (**B**) gene regions in the cp genomes of nine *Astragalus* species. Notably, the genes highlighted in red in each panel represent those which exhibited higher Pi values.

**Figure 7 cimb-47-00978-f007:**
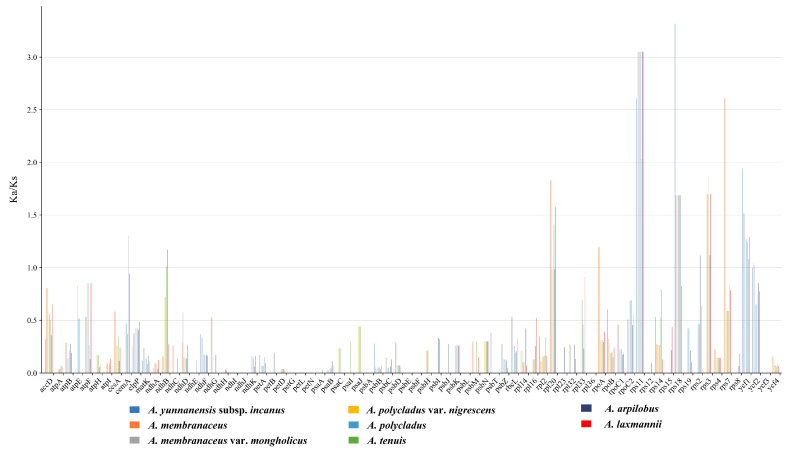
The Ka/Ks ratios of 76 PCGs from 8 *Astragalus* species taking *A. yunnanensis* as a reference.

**Figure 8 cimb-47-00978-f008:**
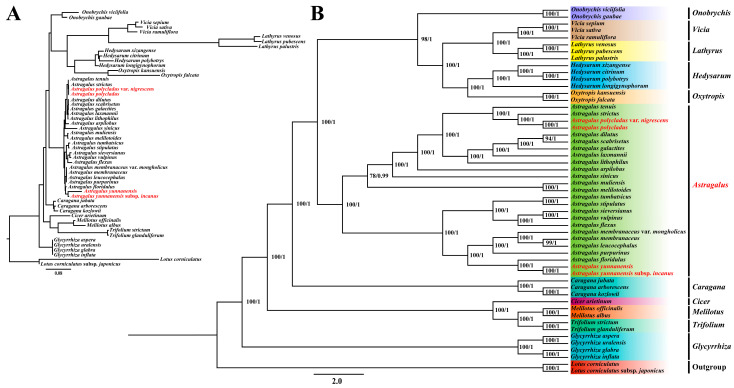
Maximum-likelihood (ML) and Bayesian inference (BI) phylogenetic tree based on the complete chloroplast genome sequence of 53 species from the Papilionaceae. (**A**) Phylogenetic tree with branches scaled to genetic distance. (**B**) Cladogram showing the topology without branch lengths. Number above nodes are support values with ML bootstrap (BS) values on the left and BI posterior probability (PP) values on the right. Species labeled in red are those sequenced in this study.

**Table 1 cimb-47-00978-t001:** The basic chloroplast genome information of 9 *Astragalus* species.

Plastome Characteristics		*A. yunnanensis*	*A. yunnanensis* Subsp. *incanus*	*A. polycladus*	*A. polycladus* var. *nigrescens*	*A. membranaceus*	*Astragalus membranaceus* var. *mongholicus*	*A. tenuis*	*A. laxmannii*	*A. arpilobus*
GenBank accession		PV156652.1	PV156653.1	PV910878	PV910879	OR528897.1	OR712437.1	OP723862.1	NC_085710.1	NC_077549.1
Protein coding genes (PCG)	Length (bp)	66,111	65,994	65,916	65,916	65,745	65,742	65,922	65,922	65,892
	GC (%)	36.53	36.54	36.43	36.44	36.53	36.52	36.42	36.43	36.36
	Length (%)	52.57	52.56	53.65	53.64	53.21	53.3	53.59	53.68	53.55
	Number	76	76	76	76	76	76	76	76	76
tRNA	Length (bp)	2268	2268	2296	2296	2066	2194	2309	1999	2269
	GC (%)	52.51	52.82	52.7	52.7	51.84	52.64	51.41	51.98	52.67
	Length (%)	1.8	1.81	1.87	1.87	1.67	1.78	1.88	1.63	1.84
	Number	30	30	30	30	27	29	30	26	30
rRNA	Length (bp)	4507	4509	4509	4509	4509	4512	4509	4509	4512
	GC (%)	54.14	54.18	54.25	54.25	54.22	54.26	54.22	54.2	54.12
	Length (%)	3.58	3.59	3.67	3.67	3.65	3.66	3.67	3.67	3.67
	Number	4	4	4	4	4	4	4	4	4
Total	Length (bp)	125,752	125,549	122,868	122,880	123,548	123,349	123,012	122,796	123,054
	Number of genes	110	110	110	110	107	109	110	106	110
	GC (%)	34.22	34.18	34.14	34.15	34.07	34.1	34.1	34.11	33.97

**Table 2 cimb-47-00978-t002:** The chloroplast-encoded genes of the 4 *Astragalus* species.

Category for Gene	Group of Genes	Name of Genes
Gene for photosynthesis	Subunits of NADH-dehydrogenase	*ndhA **, *ndhB **, *ndhC*, *ndhD*, *ndhE*, *ndhF*, *ndhG*, *ndhH*, *ndhI*, *ndhJ*, *ndhK*
	Subunits of photosystem I	*psaA*, *psaB*, *psaC*, *psaI*, *psaJ*
	Subunits of photosystem II	*psbA*, *psbB*, *psbC*, *psbD*, *psbE*, *psbF*, *psbH*, *psbI*, *psbJ*, *psbK*, *psbL*, *psbM*, *psbN*, *psbT*, *psbZ*
	Subunits of cytochrome b/f complex	*petA*, *petB* *, *petD* *, *petG*, *petL*, *petN*
	Subunits of ATP synthase	*atpA*, *atpB*, *atpE*, *atpF* *, *atpH*, *atpI*
	Large subunit of rubisco	*rbcL*
Self-replication	Small subunit of ribosome	*rps2*, *rps3*, *rps4*, *rps7*, *rps8*, *rps11*, *rps12 **, *rps14*, *rps15*, *rps18*, *rps19*
	Large subunit of ribosome	*rpl2 **, *rpl14*, *rpl16 **, *rpl20*, *rpl23*, *rpl32*, *rpl33*, *rpl36*
	DNA dependent RNA polymerase	*rpoA*, *rpoB*, *rpoC1 **, *rpoC2*
	tRNA genes	*trnA-UGC **, *trnC-GCA*, *trnD-GUC*, *trnE-UUC*, *trnF-GAA*, *trnG-GCC*, *trnG-UCC **, *trnH-GUG*, *trnI-CAU*, *trnI-GAU **, *trnK-UUU **, *trnL-CAA*, *trnL-UAA **, *trnL-UAG*, *trnM-CAU*, *trnN-GUU*, *trnP-UGG*, *trnQ-UUG*, *trnR-ACG*, *trnR-UCU*, *trnS-GCU*, *trnS-GGA*, *trnS-UGA*, *trnT-GGU*, *trnT-UGU*, *trnV-GAC*, *trnV-UAC **, *trnW-CCA*, *trnY-GUA*, *trnfM-CAU*
	rRNA genes	*rrn4.5S*, *rrn5S*, *rrn16S*, *rrn23S*
Other genes	Maturase	*matK*
	c-type cytochrom synthesis gene	*ccsA*
	Envelope membrane protein	*cemA*
	Protease	*clpP* *
	Subunit of Acetyl-CoA-carboxylase	*accD*
Genes of unknown function	Conserved hypothetical chloroplast ORF	*ycf1*, *ycf2*, *ycf3 ***, *ycf4*

Note: *: contains one intron; **: contains two introns.

## Data Availability

The data presented in this study are openly available in GenBank (NCBI, https://www.ncbi.nlm.nih.gov/, accessed on 27 October 2025) under the accession number PV156652.1 (*Astragalus yunnanensis*), PV156653.1 (*A. yunnanensis* subsp. *incanus*), PV910878.1 (*A. polycladus*), PV910879.1 (*A. polycladus* var. *nigrescens*). The associated BioProject, SRA, and BioSample numbers are PRJNA1285615, SRR34972813 (*A. yunnanensis*), SRR35000060 (*A. yunnanensis* subsp. *incanus*), SRR35022202 (*A. polycladus*), SRR35152301 (*A. polycladus* var. *nigrescens*) and SAMN49770410 (*A. yunnanensis*), SAMN49770411 (*A. yunnanensis* subsp. *incanus*), SAMN49770412 (*A. polycladus*), SAMN49770413 (*A. polycladus* var. *nigrescens*), respectively.

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
