# Peer review of "A Comprehensive Comparative Analysis and Phylogenetic Investigation of the Chloroplast Genome Sequences in Four *Astragalus* Species"

_cimb, 2025, doi:10.3390/cimb47120978_

Round 1

Reviewer 1 Report

Comments and Suggestions for Authors

I carefully reviewed the ms entitled “A comprehensive comparative analysis and phylogenetic investigation of the chloroplast genome sequences in four Astragalus species” (ID: cimb-3963433). I found it well-written and I have to suggest only some comments before publication, as follows.

It is also recommended that the authors perform a revision of the English language, removing inaccuracies and repetitions. These deficiencies can decrease the value of ms, despite it is relevant.

INTRO

L1-3: Check for English language; change “For” to “The”; remove repetition of “genus”.

RESULTS

Pag. 4: Table 1. first column, check for “Protein coding gennes”, revise with genes and add (PCG)

Pag. 8: Figure 3 caption. Check for description of Figure 3A. These are species, not kind of Astragalus. Then, each column is an amino acid, not an Astragalus species, please fix.

Pag. 11. Figure 4 caption. As in Figure 3, use brackets for A, B, C, D, E

Pag. 13. Par. 2.7, L5-7: list correctly the five major groups

Pag. 14. Fig. 8A: in the tree of panel A, the names of species are lacking, please fix

DISCUSSION

Pag. 14. L13: use italics and author abbreviations for the scientific names of Glycine max and Medicago sativa

Pag. 14. L16: did you mean genome or plastome?

Pag. 15. L17 add “species identification”

Comments on the Quality of English Language

It is also recommended that the authors perform a revision of the English language, removing inaccuracies and repetitions. These deficiencies can decrease the value of ms, despite it is relevant.

Reviewer 2 Report

Comments and Suggestions for Authors

Reviewer Report on Manuscript ID: CIMB-3963433

Title: A comprehensive comparative analysis and phylogenetic investigation of the chloroplast genome sequences in four Astragalus species

This manuscript presents a thorough comparative and phylogenetic analysis of the complete chloroplast genomes of four Astragalus taxa, augmented with five previously published plastomes. The study provides valuable genomic resources for a genus that is both taxonomically complex and economically significant. The analyses—encompassing genome structure, codon usage, repeat motifs, nucleotide diversity, and selection pressure—are comprehensive and methodologically sound. The paper is well-organized and clearly written; however, several points require clarification and minor enhancement to strengthen the manuscript's impact.

Major Comments:

  • The assertion that this work "significantly advances our understanding of chloroplast genome evolution" would be strengthened by explicitly highlighting any novel findings, such as newly identified structural rearrangements, unique mutation hotspots, or lineage-specific patterns of selection discovered in your comparative analysis.
  • Given the known structural peculiarities of IRLC (Inverted Repeat-Lacking Clade) plastomes, please comment on the steps taken to validate the assemblies and rule out potential artifacts, particularly concerning the loss of the inverted repeat regions.
  • The phylogenetic analysis robustly supports the monophyly of Astragalus. To provide a deeper evolutionary perspective, please clarify the rationale for the specific selection of the 53 taxa from the IRLC. Furthermore, the inclusion of a time-calibrated phylogenetic tree would significantly bolster the evolutionary discussions within the manuscript.
  • The discussion on the paradox between molecular conservation and morphological divergence is insightful. To further enrich this section, consider proposing a more specific hypothesis. For instance, could this disparity be linked to nuclear genomic evolution driven by environmental adaptation? Referencing known adaptive genes or genomic features in Astragalus or related genera would provide a valuable framework for this discussion.
